# Systematic improvement of neural network quantum states using Lanczos

**Hongwei Chen**[1,2,3]  **Douglas Hendry**[1]  **Phillip Weinberg**[1]  **Adrian E. Feiguin**[1]
[1]Department of Physics, Northeastern University, Boston, USA
[2]Stanford Institute for Materials and Energy Sciences, Stanford University, Stanford, USA
[3]Linac Coherent Light Source, SLAC National Accelerator Laboratory, Menlo Park, USA
`{chen.hongw, hendry.d, p.weinberg, a.feiguin}@northeastern.edu`

## Abstract

The quantum many-body problem lies at the center of the most important open challenges in condensed matter, quantum chemistry, atomic, nuclear, and high-energy physics. While quantum Monte Carlo, when applicable, remains the most powerful numerical technique capable of treating dozens or hundreds of degrees of freedom with high accuracy, it is restricted to models that are not afflicted by the infamous sign problem. A powerful alternative that has emerged in recent years is the use of neural networks as variational estimators for quantum states. In this work, we propose a symmetry-projected variational solution in the form of linear combinations of simple restricted Boltzmann machines. This construction allows one to explore states outside of the original variational manifold and increase the representation power with moderate computational effort. Besides allowing one to restore spatial symmetries, an expansion in terms of Krylov states using a Lanczos recursion offers a solution that can further improve the quantum state accuracy. We illustrate these ideas with an application to the Heisenberg $J_1 - J_2$ model on the square lattice, a paradigmatic problem under debate in condensed matter physics, and achieve state-of-the-art accuracy in the representation of the ground state.

## 1   Introduction

Understanding correlated quantum systems requires dealing with a large configuration space: datasets are comprised of all possible electronic configurations $\vec{\sigma}$ and cannot be stored in the memory of the largest supercomputer. Hence, the quantum many-body problem can be interpreted as an "extreme data science" problem [13] from an information processing perspective. In a quantum wave function, each electronic or spin configuration has an associated complex amplitude $\psi(\vec{\sigma})$ determined by solving for the eigenvectors of the Hamiltonian operator. In particular, if one is interested in the zero temperature properties of the system, the solution is given by the eigenvector with the smallest eigenvalue. Finding the exact solution of a $N$ quantum bit system with interactions requires solving for the eigenvectors of a $2^N \times 2^N$ matrix. Alternatively, one can formulate the calculation as an optimization problem in which an "energy functional" $E(\psi)$ has to minimized with respect to all the $2^N$ complex amplitudes.

Since the number of configurations $d$ grows exponentially with the number of degrees of freedom (electrons, spins), this problem quickly becomes intractable. A solution consists of "compressing' the wave function by proposing a suitable guess for the amplitudes based on some variational parameters $\vec{\alpha} = (\alpha_1, \alpha_2, \cdots, \alpha_m)$. Typically, a functional form $\psi(\vec{\sigma}) = f(\vec{\sigma}, \vec{\alpha})$ based on some physical intuition is utilized to represent the amplitude of given configuration/state $\vec{\sigma}$. The optimal parameters $\alpha_i$ are determined by solving the system of equations $\nabla_\alpha E = 0$. The objective of this solution is to achieve the lowest possible energy with a number of parameters $m \ll d$.

36th Conference on Neural Information Processing Systems (NeurIPS 2022).

Some relatively simple wave functions have enjoyed various degrees of success in the past, such as those of the Jastrow type where the amplitudes can be written as pair products $f(\vec{\sigma}, \vec{\alpha}) = \prod_{ij} U(\alpha_{ij}\sigma_i\sigma_j)$. However, in recent years we have witnessed impressive developments based on the use of neural network (NN) wave functions as variational estimators [4], which have jump-started a new vibrant field of research dubbed "quantum machine learning". Notice that the optimization of the wave function parameters now translates into the "training" of the NN by minimizing the energy function that becomes a cost function (we describe the training process below). The power of NN wave functions lies in the complex non-linear structure that provides them with remarkable expressivity to represent arbitrary complex many-body states by, at the same time, being completely agnostic to the physics.

Since restricted Boltzmann machines(RBM) were originally used as a variational ansatz for finding the ground state of the quantum many-body systems [4], there has been a growing effort to investigate other forms of neural networks, including convolutional neural networks(CNN)[9, 23], recurrent neural networks(RNN)[19], graph networks[22], transformers[25], to mention a few. Thus, neural network quantum states(NNQS) become the most appealing numerical alternative to treat quantum many body systems since they can be systematically improved by adding new layers or hidden variables, for instance. In addition to the ground state search, the application of NNQS ranges from classical simulation of quantum circuits[1, 5, 32], calculation of spectral function[17, 18], thermodynamics simulation[16, 31], and quantum tomography[44].

**Contributions**   In this work, we show how one can use a mathematically simple structure, a restricted Boltzmann machine (RBM), and yet obtain values of the ground state energy that beat all previous estimates by a range of numerical methods, including using convolutional neural networks. As we describe below, instead of increasing the number of layers or hidden variables, the solution lies on considering linear combinations of RBMs. The new wave function allows one to explore a much larger space of solutions. In particular, one can use this construction to restore spatial symmetries [40, 9, 28, 29]. In addition, we propose implementing a projection method based on a Lanczos recursion using a "Krylov basis" of RBMs obtained by sequentially applying powers of the Hamiltonian operator.

The paper is organized as follows: In Sec.2.1 we describe the quantum many-body problem in the context of the Heisenberg model; in Sec.2.2 we summarize prior attempts to study this problem using NNQS; in Sec.3 we review the basic formalism, including the structure of neural network wave functions, how to incorporate the symmetries of the problem into the quantum many-body state, and the numerical training procedure to optimize it. In Sec.4 we present results of state-of-the-art calculations for the $J_1 - J_2$ Heisenberg model on the square lattice and compare to other numerical techniques. We finally close with a summary and conclusions.

## 2   The quantum many-body problem

### 2.1   Model

In the following, we will focus on quantum spin problems where the degrees of freedom $\sigma_i$ can assume two possible values $\pm 1/2$ (or "up" and "down"). Similarly, one can think of them as generic two-level systems or "qubits". In particular, we will benchmark our methods in the context of the spin $\frac{1}{2}$ antiferromagnetic Heisenberg model with nearest and next nearest neighbor interactions, the so-called $J_1 - J_2$ model defined by the Hamiltonian:

$$\hat{H} = J_1 \sum_{\langle ij \rangle} \vec{S}_i \cdot \vec{S}_j + J_2 \sum_{\langle\langle ij \rangle\rangle} \vec{S}_i \cdot \vec{S}_j, \tag{1}$$

where $\vec{S} = (\hat{S}^x, \hat{S}^y, \hat{S}^z)$ are spin operators, the first term runs over nearest neighboring sites $\langle ij \rangle$ on a square lattice and the second term runs over next nearest pairs $\langle\langle ij \rangle\rangle$ along the diagonals of the plaquettes. For convenience, in the following, we set $J_1 = 1$ as the unit of energy. In this problem, the number of possible configurations grows as $d = 2^N$. However, the ground state wave function lies on the sector with the same number of up and down spins, constraining our search to a smaller subset of states, albeit still exponentially large.

Without the $J_2$ term, the problem can be numerically solved for hundreds of spins using quantum Monte Carlo (QMC) [38]. However, the method cannot be applied to problems with frustration since

it is noticeably affected by the infamous sign problem[24]. In our case, this is due to the presence of the $J_2$ term that makes some transition probabilities ill-defined (negative). The ground states of this model are well established in two extreme cases: at small $J_2/J_1$ the system antiferromagnetically orders with wave vector $\mathbf{q} = (\pi, \pi)$; at large $J_2/J_1$ spins prefer columnar order $\mathbf{q} = (\pi, 0), (0, \pi)$, in which they aligned antiparallel in one direction, but ferromagnetically in the other. However, in the maximally frustrated regime $J_1 \sim 0.5J_2$, the system does not display any apparent order and the nature of this spin liquid state remains controversial despite significant research efforts over the past three decades[3, 6, 11, 10, 35, 39, 41, 34, 37, 26, 20, 15, 21, 45].

Therefore, we choose this Hamiltonian for two reasons: (i) it realizes a quantum spin liquid in a parameter regime near $J_2 \sim 0.5J_1$ and (ii) conventional Monte Carlo methods fail, making the model an ideal testing ground to benchmark new techniques. Variational Monte Carlo(VMC) provides a suitable alternative that can be scaled up to large two-dimensional systems without being affected by the sign problem. The quest for relatively simple yet powerful variational states has focused on neural network states, which have shown a great deal of promise. The complexity of the problem lies in the fact that many states with similar energy have very different physical properties. Therefore, an accurate representation of the ground state becomes the key to studying the nature of the quantum phase.

## 2.2 Related work

Before the concept of NNQS became a popular new alternative for simulating many-body systems, the most successful numerical techniques to treat the 2D $J_1 - J_2$ model have been the density matrix renormalization group (DMRG)[15], VMC based on a projected fermionic ansatz[20], and tensor product states[45]. Recently, some research has focused on improving the accuracy of NNQS by using deep neural networks such as CNN[9] and group-CNN[36]. The idea of applying quantum number projection to recover the symmetries of the wave function[40, 46] has proven to be effective in improving the performance of NNQS[9, 28, 29, 36]. In addition, other alternatives that enhance the quality of the approximations consist of combining NNQS with Gutzwiller-projected fermionic wave functions[12], or pair-product wave functions[30].

## 3 Method

### 3.1 Neural Network Wave Function with symmetry

An RBM wave function takes a spin configuration – a sequence of $N$ values $\pm 1/2$ – and returns a complex coefficient corresponding to the wave function amplitude. In other words, it is a function $\psi : \{-1/2, +1/2\}^N \to \mathbb{C}$. This function is highly non-linear and is parametrized by biases $\vec{a}, \vec{b}$ and weights $W$ as:

$$\psi(\vec{\sigma}^z, \vec{a}, \vec{b}, W) = e^{\sum_{i=1}^{N} a_i \sigma_i^z} \prod_{i=1}^{M} 2 \cosh\left(\sum_{j=1}^{N} W_{ij} \sigma_j^z + b_i\right). \tag{2}$$

In this expression, the number of "hidden variables" $M$ is a tunable parameter. While RBMs have remained a simple example of a basic neural network for many decades, it was only recently that their potential as variational wave functions was appreciated [4]. In this case, unlike conventional machine learning applications, the biases and weights are complex valued.

It is possible to account for certain symmetries [27] of the problem directly within the internal mathematical structure of the RBM. In particular:

- Spin flip symmetry: If the $z$-component of the total magnetization is zero ($\sum_i \sigma_i^z = 0$), the global spin flip operation $\sigma_i^z \to -\sigma_i^z$ preserves this property. Notice that since $cosh(x)$ is an even function, we can easily restore the global flip symmetry in RBM wave function by removing the "magnetic field" terms associated to biases $\vec{a}, \vec{b}$ in Eq.(2). Thus, the RBM wave function coefficients become:

$$\psi_s(\vec{\sigma}^z, W) = \prod_{i=1}^{M} 2 \cosh\left(\sum_{j=1}^{N} W_{ij} \sigma_j^z\right). \tag{3}$$

Table 1: Character table of the $C_{4v}$ point group for square lattice.

|       | $E$ | $2C_4$ | $C_2$ | $2\sigma_v$ | $2\sigma_d$ |
|-------|-----|--------|-------|-------------|-------------|
| $A_1$ | 1   | 1      | 1     | 1           | 1           |
| $A_2$ | 1   | 1      | 1     | -1          | -1          |
| $B_1$ | 1   | -1     | 1     | 1           | -1          |
| $B_2$ | 1   | -1     | 1     | -1          | 1           |
| $E$   | 2   | 0      | -2    | 0           | 0           |

- Translational symmetry: In translationally invariant systems, the ground state of the Hamiltonian is expected to preserve the translational symmetry of the lattice. By applying the momentum projection, one can construct a variational wave function that preserved the symmetry with a well-defined momentum $\mathbf{K}$:

$$\psi_{\mathbf{K}} = \sum_{\mathbf{R}} e^{-i\mathbf{K}\cdot\mathbf{R}} \psi_s(T_{\mathbf{R}}\vec{\sigma}^z, W), \tag{4}$$

where the translation operator $T_{\mathbf{R}}$ shift all particles by a distance $\mathbf{R}$. For the 2D square lattice,

$$\mathbf{R} = m\hat{\mathbf{x}} + n\hat{\mathbf{y}}. \tag{5}$$

The operator $T_{\mathbf{R}}$ will perform a translation by $m$ steps in the $\hat{\mathbf{x}}$ direction, and $n$ steps in the $\hat{\mathbf{y}}$ direction. The coefficient $\psi_{\mathbf{K}}$ now satisfies the translational symmetry at the cost of requiring a computation time $N$ time larger.

- Lattice point symmetry: As our target model is on the 2D square lattice, the point group symmetries, which consist of rotation and reflection operations, may also be included:

$$\psi_{\mathbf{K}\mathcal{L}} = \sum_{\mathbf{R},\mathcal{L}} e^{-i\mathbf{K}\cdot\mathbf{R}} \chi(\mathcal{L}) \psi_s(T_{\mathbf{R}}\mathcal{L}\vec{\sigma}^z, W), \tag{6}$$

where $\mathcal{L}$ is the symmetry operation in the $C_{4v}$ point group, and $\chi(\mathcal{L})$ is the character of the irreducible representation $I$ for the symmetry operation $\mathcal{L}$. Since there are 8 operations in the $C_{4v}$ point group, as shown in table 1, $\psi_{\mathbf{K}\mathcal{L}}$ is 8 times more expensive to calculate compared to $\psi_{\mathbf{K}}$.

Notice that even though the computational cost of optimizing and evaluating observables with the symmetrized wave function has increased, the resulting state has a much larger expressivity than the original one, translating into a remarkable accuracy as we shall demonstrate. We should highlight here that the new states, by being linear combinations of RBMs, are no longer RBMs, and therefore allow one to explore a much larger space outside the original manifold defined by $\psi_s$, Eq.(2).

## 3.2 Wave Function Optimization

The goal of the calculation is to minimize the cost function defined by the expectation value of the energy:

$$E_{var} = \frac{\langle\psi_{\mathbf{K}\mathcal{L}}|H|\psi_{\mathbf{K}\mathcal{L}}\rangle}{\langle\psi_{\mathbf{K}\mathcal{L}}|\psi_{\mathbf{K}\mathcal{L}}\rangle} \tag{7}$$

$$= \sum_{\vec{\sigma}} P_{\vec{\sigma}} E_{loc}(\vec{\sigma}), \tag{8}$$

where the probability distribution is determined by the normalized wave function coefficients

$$P_{\vec{\sigma}} = \frac{|\langle\vec{\sigma}|\psi_{\mathbf{K}\mathcal{L}}\rangle|^2}{\sum_{\vec{\sigma}'}|\langle\vec{\sigma}'|\psi_{\mathbf{K}\mathcal{L}}\rangle|^2} \tag{9}$$

and the local energy is given by

$$E_{loc}(\vec{\sigma}) = \frac{\langle\vec{\sigma}|H|\psi_{\mathbf{K}\mathcal{L}}\rangle}{\langle\vec{\sigma}|\psi_{\mathbf{K}\mathcal{L}}\rangle}. \tag{10}$$

By formulating the problem in probabilistic terms, one can resort to Metropolis-Hastings Markov Chain Monte Carlo to evaluate the averages. The sampling over the spin configurations $\vec{\sigma}$ is carried

out by randomly flipping pairs of anti-aligned spins, and using von Neumann rejection according to a transition probability $W = |\langle \vec{\sigma}_{new} | \psi_{\mathbf{KL}} \rangle|^2 / |\langle \vec{\sigma}_{old} | \psi_{\mathbf{KL}} \rangle|^2$.

The wave function optimization can be implemented by a variety of methods. Since the energy landscape is extremely complex, simple gradient descent tends to get trapped into metastable solutions. More sophisticated strategies are usually employed, such as natural gradient descent or "stochastic reconfiguration"[42]. In contrast to the "standard" natural gradient descent method, the Fubini-study metric[33], which is the complex-valued form of Fisher information, is used to measure the "distance" between wave functions $|\psi\rangle$ and $|\phi\rangle$:

$$\gamma(\psi, \phi) = \arccos \sqrt{\frac{\langle\psi|\phi\rangle\langle\phi|\psi\rangle}{\langle\psi|\psi\rangle\langle\phi|\phi\rangle}}. \tag{11}$$

The procedure to update variational parameters using natural gradient descent is well described in literature[4, 8, 30], and we hereby summarize it. The optimization is done by minimizing the Fubini-study metric between $|e^{-d\tau H}\psi(\theta)\rangle$ and $\psi(\theta + \delta\theta)\rangle$ where $d\tau$ is a small step in imaginary time and can be viewed as learning rate in the training of neural network. The optimal choice for $\delta\theta$ is given by the solution of a system of equations:

$$\sum_{k'} \left[ \langle O_k^\dagger O_{k'} \rangle - \langle O_k^\dagger \rangle \langle O_{k'} \rangle \right] \delta\theta_{k'} = -d\tau \left[ \langle O_k^\dagger H \rangle - \langle O_k^\dagger \rangle \langle H \rangle \right], \tag{12}$$

where the log derivative $O_k = \frac{1}{\psi(\theta)} \frac{\partial \psi(\theta)}{\partial \theta}$ and $\langle \cdots \rangle$ means an average over samples. We update the parameters by $\theta_k = \theta_k' + \delta\theta_k$ and repeat until convergence is reached.

### 3.3 Lanczos recursion

Using the symmetrized RBM wave function combined with the stochastic reconfiguration method, a good approximation of the ground state can be achieved after hundreds or thousands of iterations. However, due to the limited representation power of neural network wave functions, and the errors stemming from the Monte Carlo sampling and the optimization method, the true ground state of the Hamiltonian $H$ can still differ significantly from the variational one. One possible way to increase the expressivity of the wave function is to introduce additional hidden variables or layers. However, an alternative to systematically improve the neural network wave function consists of applying a modified Lanczos recursion [14, 2, 20]. The procedure begins with a (normalized) trial wave function $\psi_0$, which in our case is an initial guess for the ground state, $\psi_0 = \psi_{\mathbf{KL}}$. Then, a new state $\psi_1$ is constructed by applying the Hamiltonian on $\psi_0$ and subtracting the projection over $\psi_0$ in order to preserve orthogonality:

$$\psi_1 = \frac{H\psi_0 - \langle H \rangle \psi_0}{(\langle H^2 \rangle - \langle H \rangle^2)^{1/2}} \tag{13}$$

where $\langle H^n \rangle = \langle \psi_0 | H^n | \psi_0 \rangle$. Notice that $\psi_1$ is orthogonal to $\psi_0$ and also normalized. In the usual Lanczos method, this recursion can be continued such that a new complete orthogonal basis can be constructed. In this representation, the Hamiltonian will have a tri-diagonal form. However, we only use $\psi_0$ and $\psi_1$ as our basis, and thus the Hamiltonian will be a $2 \times 2$ matrix.

The eigenvector $\tilde{\psi}_0$ that corresponds to the lowest eigenvalue $\tilde{E}_0$ of this matrix will be a better approximation of the true ground state of Hamiltonian compared to $\psi_0$. The lowest eigenvalue and corresponding eigenvector are

$$\tilde{E}_0 = \langle H \rangle + v\alpha, \tag{14}$$

$$\tilde{\psi}_0 = \frac{1}{(1+\alpha^2)^{1/2}} \psi_0 + \frac{\alpha}{(1+\alpha^2)^{1/2}} \psi_1, \tag{15}$$

where

$$v = (\langle H^2 \rangle - \langle H \rangle^2)^{1/2} \tag{16}$$

$$r = \frac{\langle H^3 \rangle - 3\langle H^2 \rangle \langle H \rangle + 2\langle H \rangle^3}{2(\langle H^2 \rangle - \langle H \rangle^2)^{3/2}} \tag{17}$$

$$\alpha = r - (r^2 + 1)^{1/2}, \tag{18}$$

Table 2: Ground state energy per site $E/N$ and the spin structure factor $S(\mathbf{q})$ obtained by our RBM wave function, CNN[9], RBM+PP[30], and exact diagonalization[39] for the $J_1 - J_2$ model on $6 \times 6$ square lattice. $p$ represents the number of Lanczos steps applied.

| $6 \times 6$ | $J_2 = 0.5$ | $J_2 = 0.55$ | $J_2 = 0.6$ |
|---|---|---|---|
| Energy(Exact) | -0.503810 | -0.495178 | -0.493239 |
| Energy(CNN) | -0.50185(1) | -0.49067(2) | -0.49023(1)) |
| Energy(RBM+PP) | -0.503765(1) | -0.495075(1) | - |
| Energy(RBM) | -0.50364(2) | -0.49501(1) | -0.49298(5) |
| Energy(RBM) $p = 1$ | -0.50376(3) | -0.49512(4) | -0.49313(5) |
| Energy(RBM) $p = 2$ | -0.50378(4) | -0.49514(4) | -0.49318(5) |
| $S(\pi, \pi)$(Exact) | 1.16989 | 0.89452 | 0.5545 |
| $S(\pi, \pi)$(RBM) | 1.177(8) | 0.902(6) | 0.555(4) |
| $S(\pi, 0)$(Exact) | 0.201907 | 0.2489 | 0.48412 |
| $S(\pi, 0)$(RBM) | 0.200(2) | 0.246(2) | 0.486(6) |

The eigenvector $\tilde{\psi}_0$, being a linear combination of $\psi_0$ and $\psi_1$, is the improved neural network wave function, and $\tilde{E}_0$ is the new improved variational energy. By considering $\tilde{\psi}_0$ as the new trial wave function replacing $\psi_0$, this method can be repeated to further improve the wave function. The neural network wave function obtained during the Lanczos recursion can be generalized as

$$|\Psi_p\rangle = (1 + \sum_{i=1}^{p} \beta_i H^i)|\psi_0\rangle, \tag{19}$$

where $p$ is the maximum number of Lanczos steps, and $\beta_i$ is the wave function coefficient corresponding to $H^i|\psi_0\rangle$. In this form, one can easily identify the wave function as an expansion on a Krylov basis.

In practice, taking into account the fact that the computational complexity increases dramatically with increasing $p$, only a few steps can be calculated for a large quantum many-body system. In this study, and for illustration purposes, we shall consider only the $p = 1$ or $p = 2$ cases.

## 3.4 Implementation details

In this work, we focus on the 2D $J_1 - J_2$ Heisenberg model on $L \times L$ square lattices where $L$ is an even number. For the neural network, we use $\psi_{\mathbf{K}\mathcal{L}}$ in all simulations and consider three different values for the number of hidden variables $M$ consisting of 2, 2.5, and 3 times of the number of spins $N = L^2$ in the system. The parameters $\mathbf{W}$ in the RBM are initialized to be randomly chosen random numbers with a uniform distribution between $[-0.01, 0.01]$ for both real and imaginary parts. The ground state can belong to the $A_1$ or $B_1$ irreducible representations of the $C_{4v}$ point group, depending on the value of $J_2/J_1$. In our calculations we consider both cases near the transition between the spin liquid phase and the columnar phase with $\mathbf{K} = (\pi, 0)$, i. e. for $J_2/J_1 \geq 0.5$.

Due to a large number of parameters and the numerical noise in sampling, we implement the conjugate gradient method to solve the system of equations, Eq.(12). To stabilize the method, we introduce a ridge parameter $\lambda = 10^{-6}$. For each training step, we collect 10000 samples to evaluate averages as mentioned in Sec. 3.2 including the variational energy and log derivatives. Since the adjacent states in the Markov chain are highly correlated, the number of the skipped states between samples $N_{skip}$ is chosen according to this relation $N_{skip} = 5 \times 1.0/r$, where $r$ is the acceptance rate in the previous training step. The typical value for $N_{skip}$ is from 30 to 100. As for evaluation, we collect $2 \times 10^5$ samples to calculate the average and statistical error. The learning rate used in the training ranges from $5 \times 10^{-4}$ to $3 \times 10^{-2}$. Once we observe that variational energy is not decreasing, a smaller learning rate(half of the previous one) is used instead. For large $L$, to save training time, we initialize the parameters $\mathbf{W}$ in $\psi_{\mathbf{K}\mathcal{L}}$ using the parameters trained by means of the cheaper wave function $\psi_s$. All simulations are performed using Eigen and Intel MKL on Intel E5-2680v4 and AMD Rome 7702 CPU nodes. Source code will be available at: https://github.com/hwchen2017/Lanczos_Neural_Network_Quantum_State.

Table 3: Comparison of the ground state energy per site $E/N$ and the spin structure factor $S(\mathbf{q})$ with QMC[38] results for the 2D Heisenberg model($J_2 = 0$) on $6\times6$, $8\times8$ and $10\times10$ square lattice.

| System Size | Energy(QMC) | $S(\pi,\pi)$(QMC) | Energy(RBM) | $S(\pi,\pi)$(RBM) |
|---|---|---|---|---|
| $6\times6$ | -0.678873(4) | 2.51799(6) | -0.678868(2) | 2.51(2) |
| $8\times8$ | -0.673487(4) | 3.7939(2) | -0.673482(3) | 3.79(4) |
| $10\times10$ | 0.671549(4) | 5.3124(3) | -0.671519(4) | 5.38(6) |

Table 4: Ground state energy per site $E/N$ and the spin structure factor $S(\mathbf{q})$ obtained by our RBM wave function, CNN[9], RBM+PP[30], VMC[20], and DMRG[15] for the $J_1 - J_2$ model on $10 \times 10$ square lattice. $p$ represents the number of Lanczos steps applied.

| $10 \times 10$ | $J_2 = 0.45$ | $J_2 = 0.5$ | $J_2 = 0.55$ |
|---|---|---|---|
| Energy(VMC) | -0.50811(1) | -0.49521(1) | -0.48335(1) |
| Energy(DMRG) | -0.507976 | -0.495530 | -0.485434 |
| Energy(RBM+PP) | - | -0.497629(1) | - |
| Energy(CNN) | -0.50905(1) | -0.49516(1) | -0.48277(1) |
| Energy(RBM) | -0.50916(2) | -0.49580(2) | -0.48410(3) |
| Energy(RBM) $p = 1$ | -0.5099(5) | -0.4968(4) | -0.4859(5) |
| $S(\pi,\pi)$(RBM) | 2.06(3) | 1.56(2) | 1.18(2) |
| $S(\pi,0)$(RBM) | 0.186(1) | 0.191(2) | 0.200(2) |

## 4 Results

### 4.1 Comparison with Exact Diagonalization

We benchmark the accuracy of the neural network wave functions for the ground state mainly on the $6 \times 6$ and $10 \times 10$ square lattices with periodic boundary conditions. For the $6 \times 6$ lattice, the $J_1 - J_2$ is numerically soluble by enumerating the possible spin configurations, constructing the Hamiltonian matrix, and explicitly solving the eigenvalue problem [39]. Once the ground state (or its variational approximation) is obtained, the wave function can be used to calculate other physical quantities besides the energy. Here, for illustration, we compute the spin structure factor, that defines the sublattice magnetization squared for a finite system

$$S(\mathbf{q}) = \frac{1}{N^2} \sum_{i,j} \langle \sigma_i^z \sigma_j^z \rangle e^{i\mathbf{q}\cdot(\mathbf{r_i}-\mathbf{r_j})}, \tag{20}$$

where the wave vector $\mathbf{q}$ determines spatial structure of the magnetic order. Notice that in all the tables shown here, we display the results times a factor $N$ for readability.

We first focus on the symmetrized RBM wave function without the Lanczos optimization, and we start by comparing the ground state energy for a $6 \times 6$ lattice as a function of $J_2/J_1$, as shown in Fig.1. In this figure we calculate the relative error as $|E_{nn} - E_{exact}|/|E_{exact}|$ using the exact ground state energy from Ref. [39]. We also include the relative error of the ground state energy obtained using a convolutional neural network wave function from Ref.[9]. While the relative error of the CNNs are in order of $10^{-3}$, our RBM wave function achieves an accuracy of $10^{-4}$ in the frustrated regime. Even comparing other recent works using CNNs[43, 36, 7], our RBM wave function still outperforms the CNN wave function. Besides the ground state energy, the spin structure factor computed from optimized wave functions agree very well with the exact solution as shown in Fig. 2, where the differences are smaller than the symbol size, and in data table 2.

### 4.2 Comparison with state-of-the-art quantum Monte Carlo

For larger lattices, the problem is numerically intractable. However, as mentioned before, it can be solved using QMC[38] for $J_2 = 0$. Thus, for the case without frustration we can compare with QMC results for several different lattice sizes. From table 3, we can see that even on the $10 \times 10$ lattice the energy difference is about $3 \times 10^{-5}$, showing the extraordinary accuracy of our RBM wave function.

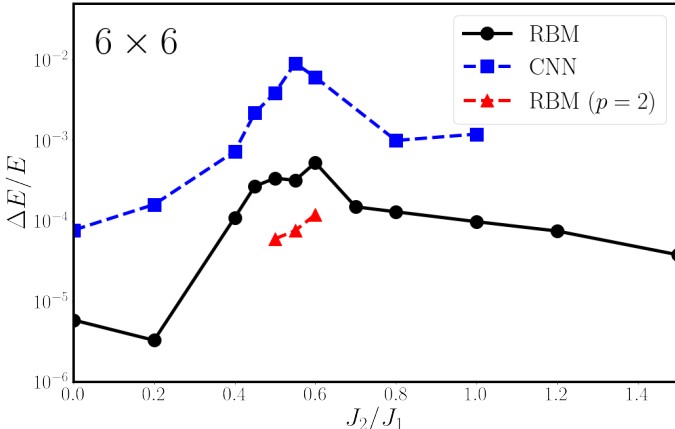

Figure 1: Relative error in the ground state energy obtained with variational Monte Carlo using symmetrized RBM wave functions (this work) and convolutional neural network (CNN) wave functions, from Ref. [9].

For the frustrated case, $J_2 \neq 0$, we compare to other methods, such as those obtained with CNN wave functions as well as results using the density matrix renormalization groump(DMRG) method with $SU(2)$ symmetry from Ref. [15] and VMC using an Abrikosov-fermion mean field with a $Z2$ gauge structure from Ref. [20]. From the data tables 4 and 5, we observe that our RBM wave function outperform the CNN wave function again in the entire range of $J_2/J_1$. In the frustrated regime, comparisons with VMC and DMRG using all the data available in literature demonstrate that the RBM wave functions still yield competitive ground state energies except at $J_2/J_1 = 0.55$ where DMRG yields a lower value.

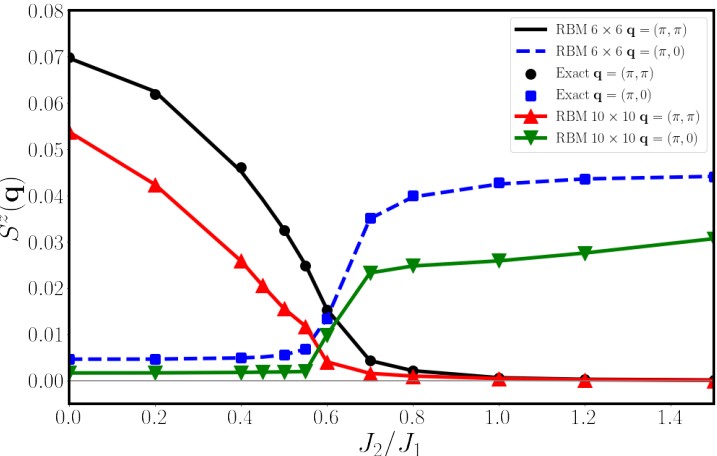

Figure 2: Static spin structure factor for ordering wave vectors $\mathbf{q} = (\pi, \pi)$ and $(\pi, 0)$ obtained with VMC using symmetrized RBM wave functions, compared to numerically exact results on a $6 \times 6$ lattice as a function of $J_2/J_1$. We also include VMC results for $10 \times 10$. Monte Carlo sampling errors are smaller than the symbol size.

### 4.3 Lanczos optimization

Since the most interesting regime lies around the maximally frustrated point $J_2 \sim 0.5J_1$, we choose 3 different values of $J_2/J_1$ using $6 \times 6$ and $10 \times 10$ lattices and perform a few Lanczos steps to further

Table 5: Ground state energy per site $E/N$ and the spin structure factor $S(\mathbf{q})$ obtained by our RBM wave function, exact diagonalization[39], and CNN[9] for the $J_1 - J_2$ model on $6 \times 6$ and $10 \times 10$ square lattice.

| $6 \times 6$ | $J_2 = 0.0$ | $J_2 = 0.2$ | $J_2 = 0.4$ | $J_2 = 0.45$ | $J_2 = 0.7$ |
|---|---|---|---|---|---|
| Energy(Exact) | -0.678872 | -0.599046 | -0.529745 | -0.51565739 | -0.530001 |
| Energy(CNN) | -0.67882(1) | -0.59895(1) | -0.52936(1) | -0.51452(1) | - |
| Energy(RBM) | -0.678868(2) | -0. 599044(3) | -0.529687(7) | -0.51552(1) | -0.529921(8) |
| S$(\pi,\pi)$(Exact) | 2.5180 | 2.22946 | 1.6604 | - | 0.15605 |
| S$(\pi,\pi)$(RBM) | 2.51(2) | 2.25(2) | 1.63(2) | 1.412(8) | 0.1567(7) |
| S$(\pi,0)$(Exact) | 0.167453 | 0.1691 | 0.17784 | - | 1.26743 |
| S$(\pi,0)$(RBM) | 0.1678(7) | 0.1681(7) | 0.1774(8) | 0.1845(8) | 1.26(2) |
| $6 \times 6$ | $J_2 = 0.8$ | $J_2 = 1.0$ | $J_2 = 1.2$ | $J_2 = 1.5$ | |
| Energy(Exact) | -0.586487 | -0.714360 | -0.848364 | -1.05268 | |
| Energy(CNN) | -0.58590(1) | -0.71351(1) | - | - | |
| Energy(RBM) | -0.586411(9) | -0.71429(1) | -0.84830(1) | -1.052640(9) | |
| S$(\pi,\pi)$(Exact) | 0.07726 | 0.02318 | 0.01115 | 0.00557 | |
| S$(\pi,\pi)$(RBM) | 0.0779(4) | 0.0233(2) | 0.0110(1) | 0.00556(5) | |
| S$(\pi,0)$(Exact) | 1.4402 | 1.54318 | 1.5729 | 1.59 | |
| S$(\pi,0)$(RBM) | 1.43(2) | 1.53(2) | 1.57(2) | 1.59(2) | |
| $10 \times 10$ | $J_2 = 0.0$ | $J_2 = 0.2$ | $J_2 = 0.4$ | $J_2 = 0.6$ | $J_2 = 0.7$ |
| Energy(CNN) | -0.67135(1) | -0.59275(1) | -0.52371(1) | -0.47604(1) | - |
| Energy(RBM) | -0.671519(4) | -0.592847(9) | -0.52388(2) | -0.47662(3) | -0.51889(2) |
| S$(\pi,\pi)$(RBM) | 5.38(6) | 4.24(5) | 2.59(4) | 0.408(3) | 0.1594(9) |
| S$(\pi,0)$(RBM) | 0.1684(9) | 0.1702(8) | 0.180(2) | 0.99(2) | 2.33(7) |
| $10 \times 10$ | $J_2 = 0.8$ | $J_2 = 1.0$ | $J_2 = 1.2$ | $J_2 = 1.5$ | |
| Energy(CNN) | -0.57383(1) | -0.69636(1) | - | - | |
| Energy(RBM) | -0.57404(2) | -0.69670(2) | -0.82565(3) | -1.02371(3) | |
| S$(\pi,\pi)$(RBM) | 0.1004(5) | 0.0480(2) | 0.0212(2) | 0.00760(6) | |
| S$(\pi,0)$(RBM) | 2.48(7) | 2.59(5) | 2.76(5) | 3.07(5) | |

improve the ground state energy. From data table 2 and 4, we see that the Lanczos steps are very effective regardless of the system size. Remarkably, by performing $p = 1$ Lanczos steps, we obtain better energy at $J_2/J_1 = 0.55$ for the $10 \times 10$ lattice that improves significantly the best available data using state-of-the-art DMRG, as shown in data table 4. Besides, compared to the "RBM+PP" results[30], which is generally considered as the start-of-the-art NNQS method, we obtain slightly lower variational energy at $J_2 = 0.5, 0.55$ on a $6 \times 6$ lattice while for a $10 \times 10$ lattice at $J_2 = 0.5$, their variational energy is $8 \times 10^{-4}$ lower than ours. Additionally, with the help of the Lanczos recursion, a better estimate of the energy can be obtained by carrying out a variance extrapolation as illustrated in Ref. [2, 20]. We also try to improve the estimation of spin structure factor using Lanczos, but the Monte Carlo sampling error makes the improvement not obvious.

## 5  Conclusion

Neural network wave functions hold a great deal of promise due to their ability to compress complex quantum many-body states within a relatively simple mathematical structure that, owing to its non-linearity, can encode an exponentially large amount of information with polynomial resources. In particular, RBM wave functions, initially deemed too simple, can be used as building blocks for systematically improved wave functions. These improved states obey the internal symmetries of the model and the point group symmetries of the lattice. In addition, they may contain contributions from the state living in a "tangent space" to the original RBM manifold. These tangent vectors are spanned in terms of powers of the Hamiltonian and form a Krylov basis.

We have demonstrated that we can achieve state-of-the-art accuracy that improves previous results using convolutional neural networks with a minimal amount of extra computational cost compared to simple RBMs. The combination of Lanczos and symmetrization offer an effective solution to problems previously beyond the reach of the most powerful numerical techniques and provide the

means to bypass the sign problem. These ideas can seamlessly translate to other areas of research ranging from materials science to quantum chemistry. Besides, our variational solution can be adopted to calculate the excitation spectrum of a quantum many-body system[17, 18], providing valuable information that can be directly compared to experiments.

**Limitations** The computational cost of a single training step scales as $\mathcal{O}(N_{sample} \times MN^2)$, where the number of hidden variables $M$ is usually proportional to the system size $N$. Thus, the computation time of calculation may be a bottleneck for its application on larger lattices. In particular, we find that even though the results for the energy are very accurate, correlation functions have relatively larger errors. This behavior might be improved by using variational forms with better representation power. Besides, the Lanczos step procedure is not size consistent, which means that the energy improvement with respect to the original wave function $|\psi_0\rangle$ vanishes for fixed $p$ and $N \to \infty$. Also, the Lanczos correction will be smaller and smaller as $p$ increases. Nevertheless, a sizable improvement is obtained even for rather large clusters with 100 sites as shown in the data table 4.

**Negative Societal Impact** Our work presents the theoretical simulation of the quantum many-body problems without any foreseeable negative societal impacts.

## Acknowledgments and Disclosure of Funding

AEF and HC acknowledge the National Science Foundation for support under grant No. DMR-2120501. DH is partially supported by a Northeastern Tier 1 grant.

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
