# Systematic improvement of neural network quantum states using Lanczos(Supplementary Material)

**Hongwei Chen**[1,2,3] **Douglas Hendry**[1] **Phillip Weinberg**[1] **Adrian E. Feiguin**[1]
[1]Department of Physics, Northeastern University, Boston, USA
[2]Stanford Institute for Materials and Energy Sciences, Stanford University, Stanford, USA
[3]Linac Coherent Light Source, SLAC National Accelerator Laboratory, Menlo Park, USA
`{chen.hongw, hendry.d, p.weinberg, a.feiguin}@northeastern.edu`

## A  Accuracy of ground state energy

To show the energy improvements from restoring different kinds of symmetry, we tried several variational wave functions $\psi_s$, $\psi_{\mathbf{K}}$, and $\psi_{\mathbf{K}\mathcal{L}}$. Their variational energies are shown in the table 1.

Table 1: The ground state energy per site for the 2D Heisenberg model($J_2 = 0$) on a $6 \times 6$, $8 \times 8$, and $10 \times 10$ square lattice obtained from the RBM wave function with different kinds of symmetries restored.

| Lattice Size | $6 \times 6$ | $8 \times 8$ | $10 \times 10$ |
|---|---|---|---|
| $\psi_s$: Spin flip symmetry(SFS) | -0.67762 | -0.67154 | -0.66737 |
| $\psi_{\mathbf{K}}$: SFS + Translational symmetry(TS) | -0.678844 | -0.673425 | -0.671288 |
| $\psi_{\mathbf{K}\mathcal{L}}$ SFS + TS + Lattice point symmetry | -0.678868 | -0.673482 | -0.671519 |

## B  Derivation of the Lanczos recursion

Using $\psi_0$ and $\psi_1$ as new basis, the Hamiltonian will be a $2 \times 2$ symmetric matrix.

$$\begin{pmatrix} \langle H \rangle & (\langle H^2 \rangle - \langle H \rangle^2)^{1/2} \\ (\langle H^2 \rangle - \langle H \rangle^2)^{1/2} & \frac{\langle H^3 \rangle - 2\langle H^2 \rangle \langle H \rangle + \langle H \rangle^3}{\langle H^2 \rangle - \langle H \rangle^2}. \end{pmatrix} \tag{1}$$

Considering the most generic $2 \times 2$ symmetric matrix $\begin{pmatrix} a & c \\ c & b \end{pmatrix}$, the two eigenvalues of this matrix can be written as :

$$\varepsilon_1 = \frac{1}{2}[a + b - ((b-a)^2 + 4c^2)^{1/2}] \quad \text{and} \quad \varepsilon_2 = \frac{1}{2}[a + b + ((b-a)^2 + 4c^2)^{1/2}]. \tag{2}$$

Plugging the expressions for $a, b, c$, we obtain Eq.(14):

$$\varepsilon_1 = \frac{1}{2}\left( \langle H \rangle + \frac{\langle H^3 \rangle - 2\langle H^2 \rangle \langle H \rangle + \langle H \rangle^3}{\langle H^2 \rangle - \langle H \rangle^2} - \left( (\frac{\langle H^3 \rangle - 3\langle H^2 \rangle \langle H \rangle + 2\langle H \rangle^3}{\langle H^2 \rangle - \langle H \rangle^2})^2 + 4(\langle H^2 \rangle - \langle H \rangle^2) \right)^{1/2} \right) \tag{3}$$

$$= \langle H \rangle + \frac{\langle H^3 \rangle - 3\langle H^2 \rangle \langle H \rangle + 2\langle H \rangle^3}{2(\langle H^2 \rangle - \langle H \rangle^2)} - (\langle H^2 \rangle - \langle H \rangle^2)^{1/2} \left( (\frac{\langle H^3 \rangle - 3\langle H^2 \rangle \langle H \rangle + 2\langle H \rangle^3}{2(\langle H^2 \rangle - \langle H \rangle^2)^{3/2}})^2 + 1 \right)^{1/2} \tag{4}$$

36th Conference on Neural Information Processing Systems (NeurIPS 2022).

$$= \langle H \rangle + (\langle H^2 \rangle - \langle H \rangle^2)^{1/2} \left[ \frac{\langle H^3 \rangle - 3\langle H^2 \rangle \langle H \rangle + 2\langle H \rangle^3}{2(\langle H^2 \rangle - \langle H \rangle^2)^{3/2}} - \left( (\frac{\langle H^3 \rangle - 3\langle H^2 \rangle \langle H \rangle + 2\langle H \rangle^3}{2(\langle H^2 \rangle - \langle H \rangle^2)^{3/2}})^2 + 1 \right)^{1/2} \right]. \quad (5)$$

The corresponding eigenvector is $\begin{pmatrix} \psi_0 \\ \alpha \psi_1 \end{pmatrix}$, with

$$\alpha = \frac{b - a - \sqrt{(b-a)^2 + 4c^2}}{2c} = \left[ \frac{\langle H^3 \rangle - 3\langle H^2 \rangle \langle H \rangle + 2\langle H \rangle^3}{2(\langle H^2 \rangle - \langle H \rangle^2)^{3/2}} - \left( (\frac{\langle H^3 \rangle - 3\langle H^2 \rangle \langle H \rangle + 2\langle H \rangle^3}{2(\langle H^2 \rangle - \langle H \rangle^2)^{3/2}})^2 + 1 \right)^{1/2} \right], \quad (6)$$

which is the explicit form of $\alpha$ in Eq.(18). Hence, the new improved eigenvector becomes

$$\tilde{\psi}_0 = \psi_0 + \alpha \psi_1. \quad (7)$$

Notice that $\tilde{\psi}_0$ is not normalized. The normalized form of $\tilde{\psi}_0$ is given by Eq.(15):

$$\tilde{\psi}_0 = \frac{1}{(1 + \alpha^2)^{1/2}} \psi_0 + \frac{\alpha}{(1 + \alpha^2)^{1/2}} \psi_1, \quad (8)$$