# OpenReview forum: "Systematic improvement of neural network quantum states using Lanczos"
_NeurIPS.cc/2022/Conference — NeurIPS 2022 Accept_

### Official Review · Reviewer_PiLy · 2022-06-13

**Rating:** 7
**Confidence:** 5
**Soundness:** 3 good
**Presentation:** 3 good
**Contribution:** 3 good

**Summary:**

The paper applies Lanczos step improvements over a shallow neural quantum state based on RBMs.
Symmetries are also exploited successfully, thanks to projections in the relevant symmetry sectors.
The authors report results that are highly competitive with the state of the art on a challenge benchmark (J1-J2 model in 2d).

**Questions:**

1. While the authors report on a comparison on the 6x6 model, it is crucial to understand what happens on the larger 10x10 model, if they want to claim new SOTA results on the J1-J2 model. The paper by Nomura and Imada in Table 2 reports the relevant energy to compare to. How does the Lanczos-step approach compare?

2. The effect of symmetries seems absolutely crucial to improve the bare RBM results. What does it happen if one imposes translation symmetries in the weights instead of summing over the group explicitly?

**Limitations:**

yes

**Strengths And Weaknesses:**

To my knowledge, this is the first application of the Lanczos-step style improvements to neural quantum states. This idea has been applied in the past to other variational states, and carries a cost that is exponential with the number of Lanczos steps.

The technique reported here allows to improve significantly on previously reported "pure" neural quantum states results.
The main limitation of the approach is the known lack of "size extensive" scaling of the Lanczos iterations, which make them ineffective for larger systems approaching in the thermodynamic limit. However, there could be cases of finite clusters where the improvement offered is still important.

My main criticism is the lack of comparison with the current best-known result on the model (more in the questions), reported in Nomura and Imada, PHYSICAL REVIEW X 11, 031034 (2021).

---

> ### Author Response · Authors · 2022-08-02
> **Response to Reviewer PiLy**
>
> Thanks for your thoughtful comments. \
> 1: Nomura's paper (Ref. [28]) is a very relevant work that we cite repeatedly. However, the paper does not include many numbers to compare. For a 10x10 lattice, they only reported energy for $J_2 = 0.5$, which is $8\times10^{-4}$ is lower than our results(See table 4). Considering other energies reported, our method is sometimes better, sometimes worse, thus comparable in accuracy.\
> 2: Carleo and Troyer attempted this in their original work (Ref. [4]). However, imposing a translational invariant W matrix (weights) reduces the expressivity of the wave function, implying that a much larger number of hidden variables is required, and it is not even clear that this would improve the performance. Henceforth, this approach has largely been abandoned in the literature.

---

> > ### Comment · Reviewer_PiLy · 2022-08-08
> > **SOTA results and cherry picking**
> >
> > I thank the authors for their replies. I think the paper is interesting and stand with my initial assessment, however the quality of the presentation has not been improved:
> >
> > Concerning my remark on ref [28], I believe that by not reporting the energy of Nomura et al (−0.497629) in their table 4, the authors are doing a bit of "cherry-picking" of the existing results, that is not up to the standards one would expect from this conference. Table 4 now contains all energies that are higher than those obtained in the presented work, however neglecting a result where better energies are obtained. The authors must (should have) include this and discuss more explicitly that the results they obtain are not state of the art, as otherwise claimed throughout the text.
> >
> > The authors also should mention in the text that the Lanczos step approach proposed here is going to improve only marginally the wave function on larger systems. It is indeed well known that in the thermodynamic limit the Lanczos corrections will become smaller and smaller.

---

> > > ### Author Response · Authors · 2022-08-09
> > > **Response to Reviewer PiLy**
> > >
> > > We appreciate the reviewer's suggestion, and we apologize for not submitting a revision in a timely manner as reading revision is optional for reviewers. We have updated the manuscript accordingly, where the reviewer can attest of the changes addressing their concerns. In the revised manuscript we added Nomura's data in Table 4 and as "RBM+PP" results for $J_2/J_1 = 0.5, 0.55$ on a 6x6 lattice in data table 2. **We also highlighted the fact in the discussion** that we obtain slightly better variational energy at $J_2/J_1 = 0.5, 0.55$ on a $6\times6$ lattice while **for a $10\times10$ lattice at $J_2/J_1 = 0.5$, their variational energy is $8\times10^{-4}$ lower than ours**. Our method yields the best results except for this particular data point. In this case, if the reviewer insists that this approach is not SOTA, we are willing to concede this point if that satisfies the reviewer, although we have different opinions. We also added comments on the limitation of the Lanczos recursion, and we hope this can address your concern. We again thank the reviewer's appreciation of our work and valuable suggestions.

---

### Official Review · Reviewer_Cg58 · 2022-06-18

**Rating:** 5
**Confidence:** 2
**Soundness:** 2 fair
**Presentation:** 3 good
**Contribution:** 3 good

**Summary:**

In the present paper, the authors introduce a method to model the wave function of quantum systems, in particular spin systems in the $J_1$-$J_2$ model. They use a restricted Boltzmann machine (RBM) and form linear combinations of them in order to incorporate certain symmetries of the investigated system. Furthermore, the authors introduce a method to determine the ground state utilizing the Lanczos recursion to iteratively improve the estimate.

They compare their method on spin lattices of various sizes to baseline methods such as Quantum Monte Carlo and a CNN based method on the basis of comparing the ground state energies and the spin structure factor obtained through the different procedures. Their method is competitive with or outperforms the baselines.

**Questions:**

* How much memory and compute do other methods use?
* How do they scale with the system size?

**Limitations:**

The authors state that the computational cost of their method scales quadratically with the system size, which limits their ability to tackle larger systems.

**Strengths And Weaknesses:**

**Strengths**

In general, the paper is concise and well written, making it pleasant to read and understand.

The authors utilized the physical knowledge about the symmetries of the system and incorporate it into the model, which is a common theme in applying machine learning models to problems in the natural sciences.

Different concepts from machine learning (RBMs), advanced statistics (Fubini-study metric), and numerics (Lanczos recursion) are combined to form a new algorithm capable to compute the ground state energy of a spin system, which is competitive or outperforms its baselines.

**Weaknesses**

I am not familiar with algorithms for estimating the ground state energy of quantum systems, so I cannot judge how big or small the improvement of this procedure over existing baselines is. However, the difference of the results is often very small, only manifesting in the fourth or fifth digit, so it is not clear to me whether this is actually significant.

The authors mention that the key for such algorithms is that they use a small amount of memory and compute. A naive implementation would use an exponential amount of memory, while this procedure has a controllable number of variables. However, they do not discuss in detail how much compute and memory competing procedures use. They only mention in the conclusion that their method is slightly more expensive than the CNN-based method while having a better accuracy, but it is unclear what this means.

**Conclusion**

Given my criticism and my lack of knowledge in the field, I am unsure whether to accept or reject this article. For me, it is important to clarify the amount of memory and compute needed by different methods, putting their respective performances in perspective. I am willing to raise my score if this concern is addressed appropriately.

---

> ### Author Response · Authors · 2022-08-02
> **Response to significance of improvement**
>
> Thanks for pointing out the concern about the significance of the "small" improvement. We hereby again explain the meaning of pursuing extremely accurate ground state calculations. The nature of the ground state in the J1-J2 model on the square lattice is one of the longest-standing open problems in condensed matter physics. The existence of a large number of competing states in a small energy window around the ground state makes it particularly challenging, and states with very similar energies may have significantly different physical properties representing different phases, such as dimer or plaquette states. As we showed in the "model" section, there have been numerous research efforts on developing numerical algorithms capable of capturing the true ground state for decades. **Thus, an incremental improvement of even one significant digit in the energy can have important consequences and help answer open questions**. Moreover, as stated by reviewers ZGgm and PiLy, we improved the previous best result significantly/by a large factor on an exceptionally difficult/challenging problem.

---

> ### Author Response · Authors · 2022-08-02
> **Response to memory and compute**
>
> Quantum Monte Carlo methods and, in particular, variational Monte Carlo, are low-memory methods. The number of double precision float point numbers required to store in memory are of the same order of variational parameters($N^2$), where N is system size. The natural gradient descent method demands much more memory usage, and it is proportional to the product of the number of samples and parameters $O(N_{sample}\times N_{parameter})$. To be more specific, for a 10x10 square lattice, the memory used in our method ranges from a few hundred megabytes to one thousand megabytes depending on the number of samples and variational parameters. As a comparison, exact diagonalization on a 6x6 lattice may use more than 10 gigabytes of memory. The memory usage of the CNN-based VMC method will have a similar dependence on samples and parameters as ours. Thus, the memory usage between different variational ansatz depends on the parameters. From Ref.[34], we use a similar number of parameters for 6x6 lattice, and three times more for 10x10 compared to the CNN method.
>
> As for the computing cost, our method scales as $O(N_{sample}\times N \times N_{parameter})$. For other VMC methods that restore translational symmetry by averaging,  the time complexity is the same as ours. Thus, the difference depends on the number of parameters used.

---

> ### Author Response · Authors · 2022-08-02
> **Response to limitation**
>
> This is a general problem with variational Monte Carlo (VMC), and it is even worse for other approaches such as tensor networks or DMRG. Regardless, we can safely say that VMC is currently one of the most promising techniques able to tackle (relatively) large system sizes of the order of hundreds of spins.

---

### Official Review · Reviewer_37nZ · 2022-07-04

**Rating:** 4
**Confidence:** 4
**Soundness:** 2 fair
**Presentation:** 3 good
**Contribution:** 2 fair

**Summary:**

This paper proposes a symmetry-projected variational solution in the form of linear combinations of simple restricted Boltzmann machines, which allows one to explore states outside of the original variational manifold and then increase representation power. Also, an expansion in terms of Krylov states using a Lanczos recursion is used to further improve quantum state accuracy. Experiments is conducted on Heisenberg J1-J2 model on square lattice.

**Questions:**

I was aware of another recent work on solving quantum many-body problem over various lattices [1]. Have authors tried the proposed methods on other types of lattices, such as triangular and honeycomb, etc? How is the performance of proposed neural network wave function on these lattices?

[1] Kochkov, Dmitrii, et al. "Learning ground states of quantum Hamiltonians with graph networks." arXiv preprint arXiv:2110.06390 (2021).


**Limitations:**

Limitation of scaling up to larger systems is well addressed. Could authors explain more on the generality of this method on different type of lattices or even random graphs?

**Strengths And Weaknesses:**

**Strengths:**
- *clarity*: This paper is overall well written and easy to follow. The method and results are clearly presented.
- *significance*: The proposed neural network quantum state can obey the internal symmetries of the quantum model and the point group symmetries of the lattice. This could make the neural network being more effective to represent quantum state by satisfying internal constraints of quantum many-body problem. Also, Heisenberg J1-J2 model has the infamous sign problem due to frustration that cannot be handled well by traditional numerical methods. So applying neural network to this problem is of great importance to quantum physics.

**Weaknesses:**
- *originality*: Based on my background and the paper itself I cannot judge if the method of this work is novel, but apparently this work misses some relevant works, such as the following:
  - *Liang, Xiao, et al. "Solving frustrated quantum many-particle models with convolutional neural networks." Physical Review B 98.10 (2018): 104426.*
  - *A. Szabó and C. Castelnovo. Neural network wave functions and the sign problem. Physical Review Research, 2(3):033075, 2020.*
  - *Kochkov, Dmitrii, et al. "Learning ground states of quantum Hamiltonians with graph networks." arXiv preprint arXiv:2110.06390 (2021).*

   And authors don’t explain the difference between the proposed method and those works (mentioned in related works) that also try to restore symmetry. And how is the performance comparison with these methods also considering symmetry, not just one CNN method.


- *quality*: Authors primarily compare the results with a specific CNN method. Could authors explain why only compare with this specific deep learning method?. Also, experiments are only performed on square lattice. Since the compared CNN method has performed so good on square lattice, the proposed method can only show very small improvements. I think authors could show the advantage of the proposed method over other methods on some other difficult quantum systems, such as several different lattices where the ground state is harder to learn, etc.

---

> ### Author Response · Authors · 2022-08-02
> **Response to comparison with other methods**
>
> We thank the reviewer for pointing out those references. We actually cite Szabo's paper and mentioned this work in section 4.1, and we will add other suggested references in the next update. Xiao's paper is a pioneering work in this area, but their performance is much worse than ours. On the 10x10 cluster $J_2=0.5$, a variational energy $-0.4736$ was obtained as compared to our result $-0.4968$. Dmitrii's paper doesn't provide specific numbers for us to compare. But from their figure 4, we can see that their accuracy is of the same magnitude as Choo's paper (Ref. [9]), which we make a full comparison with. As shown in figure 1 of our manuscript, we can conclude that our relative error is at least 10 times smaller than theirs. Since we are presenting our method and results as stat-of-the-art, we need to compare to the most accurate results available in the literature obtained by competing techniques or variational wave functions. Therefore, we compared the best QMC (Ref. [36]), VMC (Ref. [20]), and DMRG (Ref. [15]) results available. For NNQS, we select Choo's paper (Ref. [9]) and its follow-up work (Ref. [7]) because it still has the best accuracy compared with recent CNN work, and it offers lots of specific numbers to compare with. In addition, one important contribution of our work is precisely that by restoring similar symmetry, relatively simple NNQS such as RBMs can outperform more deep and complex NNQS, which is an important finding.

---

> > ### Comment · Reviewer_37nZ · 2022-08-07
> > **Response to authors**
> >
> > Thanks for the author’s reply. I understand there’s not enough time for authors to do experiments on all different lattices with various J2 values. But if possible, I still would like to see the performance of your method on 32 sites honeycomb with J2=0.2, and 36 sites kagome with J2=0. (compare with value in [1] Table IV)
> >
> > Another concern is that the authors claim the presented work to be SOTA, but it seems results of RBM+PP are often better than the presented work. In this paper, the authors only compare their results with RBM+PP on 6x6 square with J2=0.55 but don’t compare for 6x6 square with J2=0.4, J2=0.45 and 10x10 square with J2=0.5 with RBM+PP, where RBM+PP is better than this work. So I think it is misleading to claim SOTA for the presented work.

---

> > > ### Author Response · Authors · 2022-08-09
> > > **Response to Reviewer 37nZ**
> > >
> > > Thanks for bringing this data to our attention. It again shows the superiority of our method on both square and triangular lattice. But due to the approaching deadline, we don't have enough time to work on kagome and honeycomb lattices. We hope to encourage other researchers to use similar strategies on other quantum systems and lattices. In addition, adding other geometries would imply a significant re-write of the manuscript and distract the reader from the main message. As states before, we think that in order to focus on the method we focus on only one particular problem, which already is significantly challenging and relevant to the physics community.
> > >
> > > Using Lanczos recursion to systematically improve NNQS is the main contribution of our work, and we select a few $J_2$ with the highest relative error to illustrate the effect of Lanczos recursion. Therefore, it is appropriate to compare our Lanczos' results with "RBM+PP" results. **But we did not carry our Lanczos steps for $J_2=0.4, 0.45$, hence, our results for those particular data points are not representative of the accuracy of the Lanczos method**. For $J_2/J_1=0.5, 0.55$ on a 6x6 square lattice, our results are slightly better than "RBM+PP" results(See table 2). For a 10x10 lattice, they reported energy for $J_2/J_1 = 0.5$, which is $8\times10^{-4}$ is lower than our results(See table 4). Our method yields the best results except for this particular data point. In this case, if the reviewer insists that this approach is not SOTA, we are willing to concede this point though we have different opinions.  **We added these results in the data table 2 and 4 in the revision, and also highlighted the fact in the discussion that our variational energy at $J_2/J_1 = 0.5$ on a 10x10 lattice is second to "RBM+PP" results**. We hope this can address the reviewer's concern.

---

> ### Author Response · Authors · 2022-08-02
> **Response to questions about lattices**
>
> In the spin liquid region, the CNN method doesn't provide cutting-edge results as they said (Ref.[9]). In the frustrated region on a 6x6 lattice, our relative error is 10 to 100 times smaller compared to the best CNN-based NNQS method. As stated by reviewer ZGgm and PiLy, we improved the previous best result significantly/by a large factor on an exceptionally difficult/challenging problem. We thank and agree reviewer's suggestion for showing the advantage of our method by applying it to different quantum systems and different lattices. But by showing better ground state energy for the entire range of the J2/J1 parameter and specifically in the frustrated region for the $J_1-J_2$ model in 2D, which is a challenging benchmark as agreed by reviewers ZGgm and PiLy, our conclusion has firm support.
>
> Our method can be seamlessly applied to different types of lattices like triangular, honeycomb, and kagome lattices or even random graphs. The only thing that needs to be modified is the lattice point symmetry which depends on the symmetry properties of the target lattice. Due to the time limit, we try it on a 6x6 triangular lattice for two different $J_2$ values, 0 and 0.125, used in the G-CNN paper (Ref.[34]). We obtain variational energy -0.55994, -0.51415 for $J_2 =0.0, 0.125$ respectively without using Lanczos as a comparison with -0.55922 and -0.51365 reported in the G-CNN paper. Our method still performs better. We do not include the results in the current version of the paper because it will represent a significant revision that will distract the readers from the main message, and we focus on the J1-J2 model instead, as originally presented.

---

### Official Review · Reviewer_ZGgm · 2022-07-08

**Rating:** 6
**Confidence:** 4
**Soundness:** 4 excellent
**Presentation:** 3 good
**Contribution:** 3 good

**Summary:**

The paper demonstrates that by leveraging symmetrization of variational states and employing Lenczos recursion, neural network quantum states can surpass the current state-of-the-art accuracy for representing the ground state of the J1-J2 model on the square lattice. The main finding is that instead of using more intricate NN anzatz (e.g., ConvNets, Graph NN, or Transformers), the paper proposes that using the much simpler RBM anzatz coupled with symmetrization and Lanczos steps is sufficient to represent complex wave-functions. With that method they show that they can obtain much better ground state energy estimation across the entire range of the J2/J1 parameter and specifically near the point of highest frustation (~ 0.5).

**Questions:**

* On the topic of symmetries it would have been helpful for the authors to discuss other approaches to brute-force symmetrization, e.g., symmetric constructions of RBMs / ConvNets / Graph NN. It is especially interesting that applying brute-force symmetrization to RBMs worked better than the ConvNet used in [1] which follows translational symmetry by construction and uses the same brute-force symmetrization only for the C4 symmetries. It would be great if the authors could comment on why that might be the case given that both models follow the same symmetries and one (the ConvNet) is supposedly more expressive (prior to using Lanczos). Moreover, given that the set of methods advocated by the paper could be applied to any anzatz, have the authors considered using ConvNets or any other NN architecture to verify whether the improved results are due to the use of simpler RBMs, or comparing the use of brute-force symmetrization vs. inherently symmetric models (whether using symmetric RBMs, or symmetric ConvNets)?

* There is a slight mismatch between the usual meaning of invariance to symmetries and how you define your symmetrized anzatz. Specifically, one usually refers to symmetric invariant function as one where f(x) = f(Tx) for all T in some set of symmetric operators, but in this case it appears that this requirement is soften to be equal in amplitude but it does allow for a shift in phase, and so it is more akin to equivariant property. Could the authors please elaborate on this point to clarify what is the property they wish to enforce and why?

* "point group symmetries" might not be clear for all readers, and it would be great to be specific on what it means in this context (rotations + reflections).

* While not mandatory, given that it is a method defined by prior works, it would be helpful to include a derivation of equations 14-18 in the appendix to this work.

**Limitations:**

The authors have properly addressed the limitations of their approach, and the difficulty of scaling it to larger lattices. My only comment would be that the limitations paragraph at the end should also include the cost of symmetrization in the training step (so it should be N^3 not N^2), and that they should also repeat the cost of the Lencoz step. I would also add that it might be challenging to scale this method to larger NN, which might be necessary to solve certain cases with high accuracy.

**Strengths And Weaknesses:**

The paper is well written and offers a simple introduction to the subject matter that is a bit outside of the common domain of ML. The main strength of the paper are its impressive results, improving on the previous best results by a large factor on a problem that is considered exceptionally difficult. Moreover, it is noteworthy that these improvements are obtained by using relatively simple RBMs, as opposed to more intricate architectures (e.g., ConvNets or Transformers).

However, the presented methods are not novel and are based on previous works. Specifically, it appears that the symmetrization method is identical to those used by prior papers including with NN-based approaches [1,2], and to the extent of my knowledge it is a well-known widely-used technique in the field. Similarly, Lanczos iteration for improving the accuracy of a given ground-state approximation was suggested by other works and was specifically used in the past to improve results on the J1-J2 model [3]. It is worth noting that to the best of my knowledge this is the first use of this method with NN quantum states. While the authors do properly cite prior works, it is not sufficiently emphasized that they merely implement these ideas for NN quantum states -- something that should have been more clearly articulated in the introduction and abstract.

As a final note, while the paper clearly shows the superiority of their proposed method on the given task, there is not a lot of effort spent on attributing these gains to the various choices. Some abalation study (even if done only on the 6x6 case) could go a long way to clarify the contribution of each element (e.g., employing Lanczos on a non-symmetric RBM and the ConvNet model from [1], or using RBMs with a symmetric matrix to account for translation symmetry vs. brute-force symmetrization).

[1] - Choo et al., Two-dimensional frustrated J1−J2 model studied with neural network quantum states, PRB, 100:125124, Sep 2019.

[2] - Sharir et al., Deep Autoregressive Models for the Efficient Variational Simulation of Many-Body Quantum Systems, PRL 124:020503, Jan 2020.

[3] - Iqbal et al., Spin liquid nature in the Heisenberg J1-J2 triangular antiferromagnet, PRB 93:144411, Apr 2016.

---

> ### Author Response · Authors · 2022-08-02
> **Response to question 1**
>
> We thank the reviewer's insightful suggestions and comments. We do not have a clear understanding of why our RBM wave functions outperform ConvNets. Even though ConvNets are invariant by construction, Choo et al (Ref. [9]) include the rotational by averaging over them as we do. Unveiling the reasons for this is beyond the scope of our work, but this could be a novel research topic for the NNQS community.
>
> Earlier works, such as the original paper by Carleo and Troyer (Ref. [4]) have attempted inherently symmetric RBMs. However, imposing a translational invariant W matrix (weights) reduces the expressivity of the wave function, implying that a much larger number of hidden variables is required, and it is not even clear that this would improve the performance. Henceforth, this approach has largely been abandoned in the literature.
>
> We actually tried RBM wave functions without symmetry or with only translational symmetry, and apply Lanczos steps on these. But we haven't tried Lanczos on ConvNets. Based on the three different wave functions, we conclude that Lanczos works on all three and will produce a larger improvement on the poor variational wave function. Being worried that these data would distract from the main contribution of our work, we didn't report them. We will consider putting these data in the appendix in the next update. Now we attach some of our results for $J_2=0$ below to highlight the effects of restoring the symmetries.
>
>
> | Lattice Size           | 6x6    | 8x8   |10x10   |
> | :--------              |:------- | :------ | :------|
> | No symmetry            | -0.677625   | -0.671543  | -0.66737|
> | Translational symmetry   | -0.678844  | -0.673425 | -0.671288|
> | Translational symmetry + Point group symmetry  | -0.678873 | -0.673487 | -0.671519|

---

> > ### Comment · Reviewer_Cg58 · 2022-08-09
> > **Response to the authors**
> >
> > I thank the authors for their response addressing my concerns. Given the other reviews, I'm still not fully convinced by the significance of this work and, hence, will not change my score.

---

> ### Author Response · Authors · 2022-08-02
> **Response to question 2, 3, 4**
>
> 2: Technically, translational invariance in quantum mechanics actually means that f(Tx)=K(f(x)) where K is a complex phase. This means that after a symmetry operation the wave function remains invariant up to a phase. It is true that the wave function picking up a phase is an example of equivariance but in the context of this quantum mechanical problem there is no distinction and it becomes a matter of semantics. In general, our wave function is equivariant. This is explicitly accounted for by our construction, where the phase is determined by the lattice momentum and the eigenvalues of the rotation operation. This is achieved by taking appropriate linear combinations of translations and rotation of the RBM as described by our Eq.(6).
>
> 3, 4: We appreciate the suggestions, and we will fix them in the next update.

---

> ### Author Response · Authors · 2022-08-02
> **Response to limitation**
>
> We appreciate the suggestions and we tend to agree with this assessment. We have properly addressed this comment in the manuscript but we also highlight the fact that this cost is lower or comparable to other approaches.

---

### Author Response · Authors · 2022-08-02
**General comments**

We are truly grateful for the reviewer's insightful comments and suggestions. As summarized by the reviewers, by restoring symmetry for simple RBM neural network wave functions, we obtain highly impressive results that improve previous best neural network estimations by a large factor on an exceptionally difficult problem. Moreover, through the first application of Lanczos-step style improvements to neural quantum states, we bring a new approach that produces SOTA results on the famous $J_1-J_2$ model in 2D.

---

### Meta-Review · Area_Chair_PW3L · 2022-08-24

**Recommendation:** Accept
**Confidence:** Less certain

**Metareview:**

This submission proposes to apply Lanczos step improvements over a shallow neural quantum state based on restricted Boltzmann machines (RBM). The authors report highly competitive empirical results with the state of the art on one challenge benchmark: the J1-J2 model.  The reviews are mixed for this contribution: while the provided empirical results look promising, a clear and comprehensive comparison with SOTA's results was lacking in the initial submission, which was later improved during the interaction between the authors and the reviewers. We think these suggested changes should be included in the revision.  Most of the reviewers see the novelty of using Lanczos step improvements and feel this technique could be generally applicable. However, the authors should also discuss the limitation of this technique, especially the size consistency of the approach and the fact that returns will be smaller and smaller on larger lattices, explicitly in the revision.

**Award:**

No

---

### Decision · Program_Chairs · 2022-09-14

Accept